# Menopause, Ultraviolet Exposure, and Low Water Intake Potentially Interact with the Genetic Variants Related to Collagen Metabolism Involved in Skin Wrinkle Risk in Middle-Aged Women

**DOI:** 10.3390/ijerph18042044

**Published:** 2021-02-19

**Authors:** Sunmin Park, Suna Kang, Woo Jae Lee

**Affiliations:** 1Department of Food and Nutrition, Obesity/Diabetes Research Center, Hoseo University, 165 Sechul-Ri, Baebang-Yup, Asan-Si, ChungNam-Do 336-795, Korea; roypower003@naver.com; 2City Dermatologic Clinic, Daejeon 34141, Korea; wjderma@hanmail.net

**Keywords:** genetic variants, wrinkle, *EGFR*, *MMP16*, *COL17A1*, UV exposure

## Abstract

Genetic and environmental factors influence wrinkle development. We evaluated the polygenetic risk score (PRS) by pooling the selected single nucleotide polymorphisms (SNPs) from a genome-wide association study (GWAS) for wrinkles and the interaction of PRS with lifestyle factors in middle-aged women. Under the supervision of a dermatologist, the skin status of 128 women aged over 40 years old was evaluated with Mark-Vu, a skin diagnosis system. PRS was generated from the selected SNPs for wrinkle risk from the genome-wide association study. Lifestyle interactions with PRS were also evaluated for wrinkle risk. Participants in the wrinkled group were more likely to be post-menopausal, eat less fruit, take fewer vitamin supplements, exercise less, and be more tired after awakening in the morning than those in the less-wrinkled group. The PRS included *EGFR*_rs1861003, *MMP16*_rs6469206, and *COL17A1*_rs805698. Subjects with high PRS had a wrinkle risk 15.39-fold higher than those with low PRS after adjusting for covariates, and they had a 10.64-fold higher risk of a large skin pore size. Menopause, UV exposure, and water intake interacted with PRS for wrinkle risk: the participants with high PRS had a much higher incidence of wrinkle risk than those with low PRS, only among post-menopausal women and those with UV exposure. Only with low water intake did the participants with medium PRS have increased wrinkle risk. In conclusion, women aged >40 years with high PRS-related collagen metabolism may possibly avoid wrinkle risk by avoiding UV exposure by applying sunscreen, maintaining sufficient water intake, and managing estrogen deficiency.

## 1. Background

Skin wrinkles are generated as a result of the aging process and are a measure of aging. Reducing skin wrinkling, especially on the face, results in a more youthful appearance. In the modern era, not only women but also men have concerns about reducing facial wrinkles. In recent times, people have attempted to reduce wrinkles using various procedures, such as injecting botulinum toxin into the skin, lifting technologies such as light-emitting diode packs, and functional cosmetics and foods [1]. However, it is better to prevent or reduce the generation of skin wrinkles. Despite the use of surgical procedures [2], which may cause side effects, avoiding and/or managing the risk factors to affect skin wrinkling effectively prevents skin wrinkles. The modifiable risk factors for skin wrinkles are ultraviolet (UV) exposure, menopause, smoking, lean body mass loss, dryness due to dehydration, and habitual face expression [3,4]. Vitamin C and cysteine supplementation and flavonoid-containing food intake, including vegetables, meat, and dairy products, are reported to prevent and/or reduce skin wrinkles [5].

Skin acts as the static barrier to protect the inside of the body from the outside environment and is a thermal regulator to maintain thermal homeostasis [6]. Skin is composed of connective tissues containing collagen and elastin fibers and proteoglycans that lie in the epidermis and dermis [7]. When the collagen amounts, strength, and flexibility of the skin tissues decrease, skin wrinkles are generated. The aging process causes skin damage, including collagen loss and pigment deposition, by increasing reactive oxygen species (ROS), inflammation, and proteolytic enzyme activity reducing collagen amount in the skin extracellular matrix, in turn increasing wrinkle susceptibility [8,9]. In addition to chronological aging, UV exposure acutely upregulates matrix metalloproteinases (MMPs) in the degradation of collagen, which contributes to increasing the susceptibility to skin wrinkles [10]. Other environmental risk factors, including menopause, that reduce collagen and elastin contents, are also involved in increased skin wrinkles.

Increased oxidative stress in the skin due to aging or UV exposure stimulates collagen degradation by increasing ROS due to decreasing antioxidant enzymes, superoxide dismutase (SOD), and glutathione peroxidase, which causes further elevation of oxidative stress [4]. ROS activates mitogen-activated protein kinase (MAPK) to stimulate MMPs production [11]. MMPs promote collagen degradation, collagen fragmentation, elastin fiber degradation, and the loss of functionality of elastin fibers. ROS also inhibits the transforming growth factor (TGF)-β pathway, suppressing collagen synthesis [12,13]. As a result, collagen and elastin contents and skin functions are reduced, making the skin susceptible to wrinkles. The molecular processes of generating wrinkles are linked to both environmental and genetic factors. Caucasians generally have an earlier onset and more significant skin wrinkling than Asians, but increased pigmentary problems are seen more in Asians than Caucasians. There are also reports that the genetic variants that affect skin aging in Asians are different from Caucasians [14,15]. Different genetic variants can be responsible for distinct manifestations of skin aging [15]. Genetic factors play a key role in skin wrinkle generation and prevention, and they interact with environmental factors to prevent or induce skin wrinkles.

Here, we hypothesized that polygenetic risk scores (PRS) generated by pooling genetic variants associated with collagen metabolism to influence skin wrinkles would have a strong association with skin wrinkles and that the PRS would interact with environmental factors in women. This hypothesis was examined in 128 middle-aged Korean women.

## 2. Methods

### 2.1. Baseline Characteristics of Subjects

After receiving approval from the Institutional Review Board of Hoseo University to conduct a study by collecting mouth mucosal tissues and lifestyle surveys from women aged ≥40 years (IRB approval number: 1041231-181211-HR-088-02), volunteers were recruited at the City Dermatologic Clinic (Daejeon, Korea). A total of 128 women were voluntarily recruited at Daejeon City Dermatology from 1 December 2018 to 31 December 2019. Each participant provided written consent for a skin evaluation, genetic test, and the survey. Mouth mucosal tissue was collected by swabbing after rinsing the mouth with water at the City Dermatology Clinic. The volunteers completed the surveys, including age, disease status, menopause status, and other factors that might affect skin health, such as physical activity, usage of sunscreen, outdoor activity, alcohol drinking, coffee, water and fruit intake, and smoking status. After the participants washed their faces with water, their skin status was analyzed using Mark-Vu (Mark-Vu^®^; PSI PLUS, Deajeon, Korea), a skin diagnosis system [16].

### 2.2. Survey Questionnaires

The parameters that influence skin status were evaluated, and the data were used as covariates to analyze the association of the selected genetic variants with wrinkle status. Questionnaires included the skin conditions, personal health conditions, and lifestyle factors that influenced skin conditions. The questionnaires included menopause and hormone replacement therapy, physical activity indoors and outdoors, sunscreen usage, vitamin supplementation, fruit and fruit juice intake, water intake, and smoking status (Table 1). The average number of serving sizes of fruit, fruit juice, and water was recorded during the last month. Categorical variables were assigned the higher scores when harmful behaviors for skin status were conducted. For example, UV exposure was determined by asking the primary location of activity: outdoor areas with severe sun exposure, outdoor areas with little sun exposure, and indoor activity with some sun exposure or without sun exposure were counted as 4, 3, 2, and 1, respectively. The usage of sunscreen during outdoor activities was counted as 1 for always applying, 2 for applying sometimes, 3 for applying rarely, and 4 for not applying at all. Taking vitamin supplements was counted as 1 and taking no vitamin supplementation as 2.

### 2.3. Dermal Measurement

Mark-Vu (Mark-Vu^®^; PSI PLUS Co., Ltd., Suwon, Korea), a skin diagnostic imaging system, was used to analyze the skin status [16]. Before using the imaging tool, the volunteers were instructed to remove their makeup for accurate skin analysis, and a skilled expert photographed the faces of the volunteers after blocking the external light source. Mark-Vu made photos under 4 different lighting conditions, including normal light, specular light, polarized light, and UV light.

The skin health indicators were analyzed by using the four light sources as follows: normal light analyzed skin wrinkles, tone, and pore sizes of the skin; specular light analyzed reflection, freckles, and skin luster through skin flexion analysis; polarized light analyzed brown pigment and redness of the skin; UV light analyzed acne-related fungi and melanin.

When facial skin filming was complete, the Mark-Vu program divided the resulting picture into eight areas: forehead, nose, next to the right eye, next to left eye, under the right eye and left eye, right cheek, and left cheek, and calculated four or six areas suitable for analyzing the skin health indicators. The results of each skin health indicator were assigned 1 to 3 points, compared to a database based on the age and gender of 200 people in Korea, as developed by Mark-Vu. Each score indicates the level of care or treatment needed: 1 point indicates preventive care is needed, 2 points indicates a need for ongoing maintenance, 3 points indicates a need for intensive management, and 4 points or higher scores indicates a dangerous level of skin injury.

Fitzpatrick classification of the skin type was analyzed based on skin tone (skin brightness) results and susceptibility to tanning from UV taken with regular light on Mark-Vu devices. The skin tone was scored from 1 (white, easily burns) to 6 (dark brown and easily tans). In the Fitzpatrick skin type, Koreans were mainly classified into score 3 (tans gradually and uniformly light brown), score 4 (always tans well and moderate brown), or score 5 (rarely burns, easily tans, and dark brown) [17,18]. The susceptibility to tans from UV was scored from 1 (always burns and never tans) to 6 (no burn, tans very easily). From the Mark-Vu measurements, participants with skin tone results were divided into those with 1–2, 3–4, and >4 points, classified as having Fitzpatrick skin type 3, 4, and 5, respectively.

### 2.4. Genotyping, Single Nucleotide Polymorphism (SNP) Imputation, and Quality Control

DNA samples from the volunteers were isolated from their oral mucosal cells, and genotyping was conducted using the Teragen Affimetry chip that was customized from the Affymetrix Genome-Wide Human SNP array 5.0 of Affymetrix Ltd. (Santa Clara, CA, USA). The genotypes included had high imputation quality (proper info >0.5) [19]. The accuracy of the genotyping was checked using the Bayesian Robust Linear Modeling with Mahalanobis Distance genotyping algorithm. The exclusion criteria for genotyping data were as follows: low genotyping accuracies (<98%), high missing genotype call rates (≥4%), high heterozygosity (>30%), and gender bias.

### 2.5. GWAS for Wrinkles and Construction of PRS

The wrinkle scores of 0, 1, and 2 from the Mark-Vu were categorized into 2 groups: those with scores of 0 and 1 were considered as having a normal wrinkle-free status (control) and those with a score of 2 as the wrinkled group (case). After adjusting for covariates (age, sunscreen application, and UV exposure), GWAS was conducted with the two groups categorized according to facial wrinkle status using PLINK.

SNPs were excluded for the final genetic model if they did not meet the following criteria: >0.2 minor allele frequency (MAF) and *p* > 0.05 for the Hardy–Weinberg equilibrium (HWE). Since any single SNP had a low power to explain the genetic impact on facial wrinkles, PRS was calculated from three selected genetic variants from the GWAS by adding the number of risk alleles of each genetic variant. The PRS was divided into 3 groups: 0–1 (low PRS), 2–3 (medium PRS), and ≥4 (high PRS). The PRS explained the genetic impact on wrinkle status.

### 2.6. Statistical Analysis

This study used PLINK version 2.0 (http://pngu.mgh.harvard.edu/purcell/plink, accessed on 13 September 2020) and SAS software version 7 (SAS Institute, Cary, NC, USA) for statistical analysis. The G power calculator was used to examine the sample size required [20]: a sample size of 128 was sufficient to achieve significance at α = 0.05 and β = 0.15 in the logistic analysis. Descriptive statistics of continuous and categorical variables are provided as arithmetic means with standard deviations (SDs) and frequency distributions. Statistical analysis was conducted by one-way analysis of variance and chi-square tests for continuous and categorical variables, respectively. Multivariate logistic regression was used to calculate odds ratios (OR) and 95% confidence intervals (CI) for the risk of wrinkle and other factors to explain the skin status while controlling for covariates including age, UV exposure, sunscreen usage, menopause, sleep deprivation, morning tiredness, exercise, smoking, alcohol drinking, disease status, water intake, and vitamin supplementation. To examine the interaction between the PRS groups and lifestyles influencing wrinkle status, multiple regression models, including the main effect of PRS and interaction terms of PRS and lifestyles, were used after adjusting for potential confounders. When low and high groups of lifestyles were divided for the interaction analysis with PRS, the high group was assigned as such according to the 75th percentiles of each parameter. The general linear model was used to obtain the OR and 95% CI for the wrinkle risk associated with the PRS group based on the lifestyles while controlling for the covariates.

## 3. Results

### 3.1. Demographical Characteristics and Associations with Wrinkle Risk

Sixty percent of the female participants had more wrinkles than the age-adjusted standards that Mark-Vu provided, and they were considered the wrinkled group (Table 2). The participants in the wrinkled group were older than those in the less-wrinkled group after adjusting for covariates associated with wrinkle generation. Along with age, the percentage of the participants having menopause without hormone replacement therapy was much higher in the wrinkled group (57.6%) than the less-wrinkled group (20.0%) (Table 2). However, the adjusted OR for age and menopause had no significant association with wrinkle risk after adjusting for the assigned covariates. Regarding Fitzpatrick skin type, the persons with class 5 belonged to the wrinkled group more than the less-wrinkled group (*p* < 0.01; Table 2). The adjusted ORs for Fitzpatrick skin type had a positive association with wrinkle risk after adjusting covariates by 6.526 times (Table 2). The percentage of participants with UV exposure did not differ between the wrinkled and less-wrinkled groups, but there were fewer sunscreen-using participants in the wrinkled group than the less-wrinkled group. Adjusted ORs for UV exposure and sunscreen usage were not associated with wrinkle risk (Table 2). Water, coffee, and alcohol intake were not significantly different between the two groups and they did not have a significant association with wrinkle risk. Smoking and sleep deprivation were not significantly different between the less-wrinkled and wrinkled groups. However, the number of participants with tiredness after awakening from sleep was much higher in the wrinkled group than the less-wrinkled group (Table 2). Interestingly, fruit intake, vitamin supplementation, and exercise were much lower in the wrinkled group than the less-wrinkled group (*p* < 0.05). Fruit intake decreased the wrinkle risk by 0.062-fold compared to the less-wrinkled group (Table 2). Vitamin supplementation decreased the wrinkle risk by 0.056-fold compared to those who did not supplement with vitamins.

### 3.2. Characteristics of the Three Selected Genetic Variants Related to Wrinkle Risk

Three SNPs were selected from the GWAS that were associated with wrinkles in the 128 participants. Genetic variants were selected from the genes related to wrinkle generation-related pathways with high statistical significance in GWAS. The selected SNPs were epidermal growth factor receptor (*EGFR*) rs1861003, matrix metallopeptidase-16 (*MMP16*) rs6469206, and collagen type XVII alpha 1 chain (*COL17A1*) rs805698 (Table 3). These SNPs indicated that wrinkle generation might be linked to collagen synthesis and degradation. The minor alleles of *EGFR* rs1861003 and *MMP16* rs6469206 were negatively associated with wrinkle risk by 0.2818- and 0.4812-fold, respectively, after adjusting for wrinkle risk factors, whereas those of *COL17A1* rs805698 had a positive association by 11.49-fold. The three SNPs met the criteria of MAF (≥0.05%) and Hardy–Weinberg equilibrium (HWE; *p* > 0.05) criteria (Table 3).

Wrinkle risk was also associated with genetic variants related to oxidative stress and inflammation (Table 3). *SOD3* rs4697073 related to oxidative stress had a negative association with wrinkle risk (OR = 0.206 and 0.276; *p* < 0.05). The inflammation-related genetic variants, tumor necrosis factor (TNF) receptor superfamily member (*TNFRSF*) 6b rs1291206, and nitric oxide synase-1 (*NOS1*) rs9658535 had a significantly positive association with wrinkle risk by 2.76- and 2.81-fold, whereas *TNFRSF8* rs6690493 had a significantly negative association with wrinkle risk by 0.189-fold (Table 3).

### 3.3. Adjusted ORs for Wrinkle Risk According to the PRSs Generated by Three Selected Collagen Metabolism-Related SNPs

There were no significant differences among PRS groups in demographic variables and food intake (Table 4). Demographic variables and food intake did not have a direct impact on PRS to influence wrinkles. Since each selected SNP exhibited a weak association, three selected genetic variants were pooled using PRS. High PRS indicated a genetically higher risk of wrinkles. The wrinkle risk was 15.39-fold higher in the high-PRS group than in the low-PRS group after adjusting for covariates of age, UV exposure, sunscreen usage, menopause, sleep deprivation, morning tiredness, exercise, smoking, alcohol drinking, disease status, water intake, and vitamin supplementation (Table 5). PRS did not have any significant association with other skin status markers, such as refraction, melanin deposition, browning color, blushing, and gloss. These factors that account for skin status did not show any association with PRS. However, pore size was also associated with PRS: the participants with high PRS had a 10.64-fold higher likelihood of large pore size than those with low PRS (Table 5).

Wrinkling was strongly correlated with pore size (r^2^ = 0.609, *p* < 0.001; Table 6). However, melanin deposition and brown skin color also showed a moderate correlation with wrinkles (r^2^ = 0.398 and 0.285; Table 6). Blushing, gloss, and reflection of skin surface did not have any correlation with wrinkles. These results indicated that the risks of wrinkle and pore size are related to the genes regulating collagen contents in the skin tissues.

### 3.4. Interaction of PRS and Lifestyles for Wrinkle Risk

Wrinkle risk exhibited a significant interaction with menopause (*p* = 0.005), UV exposure (*p* = 0.015), and water intake (*p* = 0.003) in two-way ANOVA with interaction terms and covariates. However, there was no significant interaction between PRS and other factors, including age and food intake. High PRS increased the percentage having high wrinkles much more than low PRS, but only in the participants with menopause (Figure 1A). High PRS increased the chance of having wrinkles compared to low PRS more in the participants with high UV exposure than those with low UV exposure (Figure 1B). Water intake interacted with PRS to influence wrinkle risk. However, in both low- and high-water-intake subjects, there was a higher percentage of participants with a high PRS than those with a low PRS, but a higher percentage of the participants with a medium PRS had wrinkles in the low-water-intake group (Figure 1C).

## 4. Discussion

Although wrinkles are a natural aging process, people want to avoid them, especially on the face. The known risk factors for wrinkles are UV exposure, age, smoking, regular facial movements, and skin dryness. Genetic traits also influence skin wrinkles. Asians and Caucasians have different skin conditions, including epidermis thickness and elasticity, and skin status affects skin wrinkles and aging [21,22]. Epidermis thickness is associated with collagen contents, and aging reduces collagen contents by increasing its degradation and decreasing synthesis, leading to the fragmentation of skin fibers [23]. This process elevates the susceptibility to skin wrinkles. The genes associated with this process may contribute to the risk of skin wrinkles. The present study demonstrated that the PRS related to collagen metabolism was associated with skin wrinkles in middle-aged Asian women and that the PRS had interactions with menopause, UV exposure, and water intake. Therefore, women with high PRS may be recommended to have less UV exposure and to mitigate the menopausal impact on wrinkle generation. To the best of our knowledge, this is the first study to show the environmental interaction with PRS, and the results may be applied to personalized skin management.

Skin is composed of the epidermis and the dermis. The epidermis is primarily made of keratinocytes, and the dermis is composed of collagenous extracellular matrix [6]. The dermis provides mechanical strength, resiliency, and elasticity to the skin. The shallow skin wrinkles are made in the epidermis, and they may be reversible since the epidermis has self-renewing capability [6]. However, the skin wrinkles are generated mainly in the dermis, partly by reducing collagen and elastin fibers [6]. Aging disturbs the dermis structure by diminishing collagen contents and fragmentation. Skin wrinkles are associated with decreased collagen contents related to its synthesis and degradation. In the skin, collagen is synthesized by EGFR activation via epidermal growth factor to stimulate the MAPK signaling pathway that may increase collagen synthesis and reduce the activities of MMPs involved in collagen degradation [24]. The downstream pathways are connected to nuclear factor-kappa B and tumor necrosis factor-α, indices of inflammation, and TGF-β pathways [25]. EGFR pathways modulated by activators such as retinol and inhibitors such as decorin influence collagen synthesis, degradation, and inflammation [24,25]. Therefore, the modulation of *EGFR* and MMP activities by changing genetic variants may influence collagen contents in the skin and affect skin wrinkles. A recent study has demonstrated that *COL17A1* expression is associated with the epidermal stem cell activation, and the eventual *COL17A1* loss in the stem cells limits their activation [26]. The maintenance of *COL17A1* may rescue skin aging markers such as wrinkles. Genetic variants of collagen metabolism may affect collagen degradation. Therefore, the selected genetic variants, *EGFR* rs1861003, *MMP16* rs6469206, and *COL17A1* rs805698, might be largely involved in regulating the collagen contents in the epidermis and dermis. In a Chinese female population, 10q26.13 (rs4962295) was associated with wrinkles under the eyes [14]. However, the genetic location has not yet been studied for skin wrinkles, but it is involved in *FGFR2*, possibly associated with skin wrinkles. Another Chinese study demonstrates that rs11979919, 3 kb downstream of *COL1A2*, shows an association with eyelid laxity [15]. These results suggested that skin wrinkles might be associated with the genetic variants involved in collagen metabolism.

The present study exhibited some risk factors for skin wrinkles: Menopause, lower fruit intake and vitamin supplementation, less exercise, and lower sleep quality were potentially risk factors for skin wrinkles in middle-aged women. UV exposure also showed increased frequencies in the wrinkled group, and sunscreen application reduced the wrinkle frequencies in the wrinkled group compared to the less-wrinkled group. However, UV exposure and sunscreen did not reach statistical significance. These risk factors found in the present study were similar to those in previous studies [27,28,29,30]. Serum 17β-estradiol concentration is an intrinsic contributor to skin wrinkles [31]. Menopause accelerates skin changes, including wrinkles, dryness, and atrophy [27,31]. However, sunscreen usage has been reported to exert adverse effects, and a paradigm shift has been suggested of switching to reducing oxidative stress instead of UV exposure and other environmental factors by using antioxidants topically in a systematic manner [28,32]. Interestingly, the present study demonstrated that fruit intake and vitamin supplementation showed a significant reduction in wrinkle frequencies. Fruit and vitamin supplements contain antioxidants, especially vitamin C, which is well-known to stimulate collagen synthesis to maintain skin collagen and thus protect against skin wrinkles [5]. In the present study, *SOD3* rs4697073, related to an antioxidant enzyme, was associated with wrinkle risk, and *TNFRSF6B* rs1291206, *TNFRSF8* rs6690493, and *NOS1* rs9658535, involved in inflammation, were associated with wrinkle risk. This suggests that oxidative stress and inflammation may be involved in wrinkle risk. These results suggested that fruit and vitamin intakes might reduce oxidative stress and inflammation, and they might show a more significant negative association with skin wrinkles than sunscreen usage. Therefore, reducing oxidative stress and inflammation may play a critical role in preventing and reducing wrinkles in the skin, possibly more important than using sunscreen.

Although UV exposure and water intake did not show significant associations with skin wrinkles in all the participants, they interacted with PRS for wrinkle risk in the middle-aged women in the present study. It suggested that genetic impact had an interaction with environmental factors for wrinkle risk. Women with high PRS were recommended to lessen UV exposure by sunscreen application and increase antioxidant intake to reduce wrinkle risk. Moreover, sufficient water intake (>2 cups/day) might prevent wrinkle risk in women with high PRS. Caution is needed in interpreting the effect of water intake on the risk of wrinkles, since water intake is not a direct measurement of skin hydration in the present study. The direct impact of water intake on skin wrinkles cannot be linked to skin hydration related to fluid retention. Menopause had a strong positive association with wrinkle risk without genetic impact. Menopause may be managed by hormonal replacement therapy, but hormone replacement therapy is reported to have the adverse effects of increased risk of breast cancer [33,34,35]. It may be helpful to manage menopausal symptoms by using phytoestrogen treatments instead of hormonal replacement therapy. Therefore, women with high PRS may be advised to avoid UV exposure and to have sufficient water and phytoestrogen intake. The interaction of genetic factors related to collagen metabolism with some environmental factors was a novel finding of the present study.

This study had some limitations. The sample size may not have been large enough to show significant effects for some parameters. Furthermore, the study design was a case-control study that did not show a cause-and-effect relationship; however, the data collections were well-organized and controlled by the nurses and dermatologist. The study scope was limited to middle-aged women since the sample size was small for a genetic study, and men and women may be differently affected by genetic and lifestyle factors for wrinkle generation. Therefore, the study can be reliable, although the sample size was not very large. However, the association between genetic variants related to collagen metabolism and wrinkle risk may be further elucidated in a large cohort.

## 5. Conclusions

The PRSs of *EGFR* rs1861003, *MMP16* rs6469206, and *COL17A1* rs805698 related to collagen metabolism was positively associated with skin wrinkle risk in middle-aged Korean women. It might also be related to Fitzpatrick skin type and pore size on the face, suggesting that collagen contents in the skin might be involved with wrinkles and pore size. Menopause, UV exposure, and water intake had interactions with PRS for wrinkle risk: the participants with high PRS had a much higher wrinkle risk than those with low PRS, especially with menopause and UV exposure. Women with high PRS might be recommended to reduce UV exposure by applying sunscreen, drink sufficient water, and manage menopause to avoid wrinkle risk. The results of this study may help develop future guidelines for genotype-based personalized skin management.

## Figures and Tables

**Figure 1 ijerph-18-02044-f001:**
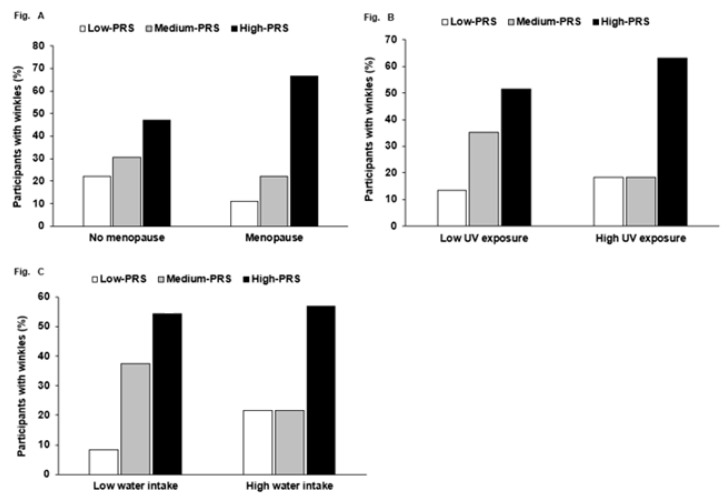
The frequency distribution of participants having wrinkle risk in the three risk groups: polygenetic risk scores (PRSs) according to menopause, UV exposure, and water intake. PRS was calculated by summing the number of risk alleles of three selected SNPs from a genome-wide association study (GWAS) with wrinkle risk, and PRSs were categorized into three groups by the tertiles (low PRS, medium PRS, and high PRS). PRS had an interaction with menopause, UV exposure, and water intake. The incidence (%) of high wrinkling in the three PRS groups in (**A**) no menopause and menopause, (**B**) low and high UV exposure (cutoff: three scores of the places held in significant activity during the day), (**C**) and low and high water intake (cutoff: 2 cups/day).

**Table 1 ijerph-18-02044-t001:** Survey questionnaires influencing skin wrinkles.

Question Category	Main Questions	Answers
Personal condition	Age	( )
	Sun exposure	( ) hours
	Having metabolic diseases	Yes or no
	Menopause	No, yes with HRT treatment, or yes without treatment
	Sleep	Normal or cannot sleep because of mental stress
	Tiredness	Normal or tired
	Exercise	Outside, inside, or none
	UV exposure	Less than 1 h/day, 1~3 h/day, or >3 h/day
	Sunscreen usage	Always, sometimes, rarely, or not at all
Food and water intake	Vitamin supplementation	Fatigue recovery: yes, no, or not sure
	Fruit or fruit juice intake	Every day, occasionally (1–2 times/week), or rarely
	Fried food intake	Every day, occasionally (1–2 times/week), or rarely
	Water intake	( ) glasses ^1^ per day
	Coffee intake	( ) cups ^1^ per day
	Fast food intake	Frequently, occasionally (1–2 times/week), or rarely

^1^ Glasses and cups were indicated as 200 mL. HRT, hormone replacement therapy.

**Table 2 ijerph-18-02044-t002:** Adjusted means and odds ratio (ORs) of demographic characteristics of the participants according to facial wrinkles.

	Less-Wrinkled (*n* = 51)	Wrinkled (*n* = 77)	Adjusted ORs (95% CI) ^c^
Age (years)	52.4 ± 7.0 ^a^	56.3 ± 9.2 *	4.221 (0.345–10.01) ^d^
Fitzpatrick skin type (scores)			
3	13 (25.4) ^b^	11 (14.3)	1
4	35 (68.6)	42 (54.5)	1.166 (0.364–3.741)
5	3 (5.88)	24 (31.3) **	6.526 (1.093–38.95) *
Menopause (Yes, %)	10 (19.6)	44 (57.1) ***	1.285 (0.268–6.163) ^e^
UV exposure (Yes, %)	18 (35.3)	32 (41.6)	1.253 (0.400–3.928) ^f^
Sunscreen application (No, %)	33 (64.7)	33 (42.9) *	3.806 (0.297–48.72) ^g^
Water intake (cups/day)	3.51 ± 2.04	3.59 ± 2.07	0.629 (0.216–1.831) ^h^
Coffee intake (cups/day)	1.10 ± 1.22	1.03 ± 1.41	0.861 (0.292–2.536) ^i^
Alcohol intake (Yes, %)	20 (39.2)	37 (48.1)	1.163 (0.382–3.542) ^j^
Fruit intake (serving/day)	2.06 ± 0.63	1.72 ± 0.64 *	0.062 (0.001–0.772) ^k,^*
Vitamin supplementation (Yes, %)	43 (84.3)	43 (55.8) **	0.056 (0.007–0.428) ^l,^***
Smoking (Yes, %)	2 (3.92)	5 (6.49)	1.770 (0.062–50.6) ^m^
Exercise (Yes, %)	37 (72.5)	37 (48.1) *	0.540 (0.183–1.596) ^n^
Sleep deprivation (Yes, %)	33 (64.7)	57 (74.0)	1.375 (0.368–5.135) ^o^
Tiredness after awakening (Yes, %)	12 (29.4)	40 (51.9) *	2.314 (0.668–8.012) ^p^

^a^ Values represent adjusted means ± standard deviations or ^b^ number of subjects (percentage of each group). ^c^ Adjusted odds ratio (ORs) and 95% confidence intervals (CI). Adjusted means and ORs were calculated after adjusting for age, menopause, UV exposure, sunscreen application, tiredness after awakening, sleep deprivation, vitamin supplementation, exercise, smoking, alcohol drinking, water intake, fruits intake, and coffee intake. The cutoff points were as following: ≥60 years old for age ^d^, menopause without hormonal replacement therapy ^e^, UV exposure ^f^, sunscreen application always or often ^g^, >2 cups of water per day ^h^, ≥1 cups of coffee per day ^i^, more than 2–3 times per week of alcohol intake ^j^, >2 servings of fruit or fruit juice per day ^k^, no vitamin or functional food supplementation ^l^, cigarette smoking ^m^, indoor or outdoor exercise 2–3 times per week ^n^, often having sleep deprivation due to stress, ^o^ and feeling tiredness after awakening ^p^. * Significant differences in wrinkles at *p* < 0.05, ** *p* < 0.01, *** *p* < 0.001.

**Table 3 ijerph-18-02044-t003:** Characteristics of genetic variants that influence skin-wrinkle risk selected by genome-wide association study.

Chr ^a^	SNP	Position	Mi ^b^	Ma ^c^	ORs (95% CI) ^d^	*p*-Value for ORs ^e^	MAF ^f^	*p*-Value for HWE ^g^	Gene	Feature
7	rs1861003	55184849	G	A	0.282 (0.083–0.952)	4.14 × 10^−2^	0.1078	0.599	*EGFR*	intron
8	rs6469206	88072463	G	T	0.481 (0.244–0.949)	3.48 × 10^−2^	0.4752	0.4251	*MMP16*	intron
10	rs805698	104057158	T	C	11.49 (2.225–76.08)	4.40 × 10^−3^	0.1127	0.3557	*COL17A1*	missense
4	rs4697073	24799121	A	G	0.206 (0.045–0.949)	4.27 × 10^−2^	0.0546	1	*SOD3*	intron
20	rs1291206	63697816	A	G	2.76 (1.34–5.66)	5.71 × 10^−3^	0.3614	0.5168	*TNFRSF6B*	intron
1	rs6690493	12075469	G	A	0.189 (0.064–0.553)	2.38 × 10^−3^	0.2108	0.2295	*TNFRSF8*	intron
12	rs9658535	117219805	C	T	3.81 (1.42–10.2)	7.74 × 10^−3^	0.1765	0.7314	*NOS1*	intron

^a^ The position of the single nucleotide polymorphism (SNP) in the chromosome. ^b^ Minor allele; ^c^ major allele; ^d^ odds ratio for each genetic variant to influence wrinkle risk; ^e^
*p* value for odds ratios that have an association with wrinkle risk after adjusting for covariates including age and UV exposure. ^f^ minor allele frequency; ^g^ Hardy-Weinberg equilibrium. *EGFR*, epidermal growth factor receptor; *MMP16*, matrix metalloproteinase; *COL17A1*, collagen type XVII alpha 1 chain; *SOD3*, superoxide dismutase 3; *TNFRSF*, tumor necrosis factor (TNF) receptor superfamily member; *NOS1*, nitric oxide synthase-1.

**Table 4 ijerph-18-02044-t004:** Adjusted means of demographic characteristics of the participants according to polygenetic risk scores (PRSs) related to collagen metabolism.

	Low PRS ^a^ (*n* = 25)	Medium PRS (*n* = 48)	High PRS (*n* = 55)
Age (yrs)	52.4 ± 2.27	50.3 ± 1.61	53.1 ± 1.50
Menopause (Yes, %)	13 (52.0)	19 (39.6)	23 (41.8)
UV exposure (Yes, %)	9 (36.0)	7 (35.4)	24 (43.6)
Sunscreen usage (No, %)	16 (64.0)	31 (64.6)	31 (56.4)
Water intake (cups/day)	2.89 ± 0.56	3.59 ± 0.40	3.87 ± 0.37
Coffee intake (cups/day)	1.06 ± 0.36	0.95 ± 0.25	1.26 ± 0.23
Alcohol intake (Yes, %)	13 (52.0)	29 (60.4)	29 (52.7)
Fruit intake (serving/day)	1.93 ± 0.17	0.95 ± 0.25	1.26 ± 0.23
Vitamin supplementation (Yes, %)	20 (80.0)	31 (64.6)	36 (65.5)
Smoking (Yes, %)	1 (4.0)	2 (4.2)	2 (3.6)
Exercise (Yes, %)	9 (36.0)	29 (60.4)	37 (67.3)
Sleep deprivation (Yes, %)	5 (20.0)	18 (37.5)	25 (45.5)
Tiredness after awakening (Yes, %)	19 (76.0)	27 (56.3)	44 (80.0)

The values represent adjusted means ± standard deviations or the number of subjects (percentage of each group). PRS was generated by summing the number of the risk alleles in the collagen type XVII alpha 1 chain (*COL17A1*) rs805698, epidermal growth factor receptor (*EGFR*) rs1861003, and matrix metalloproteinase 16 (*MMP16*) rs6469206. ^a^ PRS ≤ 1, low PRS; 2–3 PRS, medium PRS; PRS ≥ 4, high PRS.

**Table 5 ijerph-18-02044-t005:** Adjusted odds ratios and 95% confidence intervals for skin characteristics according to polygenetic risk scores (PRSs) related to collagen metabolism.

	Low PRS ^a^ (*n* = 25)	Medium PRS (*n* = 48)	High PRS (*n* = 55)
Wrinkles	1 ^b^	1.037 (0.175–6.153) ^c^	15.39 (2.480–120.5) ***
Refraction of the skin surface	1	0.193 (00035–1.077)	0.668 (0.144–3.088)
Melanin deposition	1	0.690 (0.138–3.460)	1.178 (0.242–5.735)
Browning color	1	5.376 (0.266–108.5)	2.781 (0.154–50.22)
Blushing	1	0.482 (0.099–2.336)	1.106 (0.230–5.328)
Gloss	1	7.473 (0.994–55.63)	5.060 (0.617–41.51)
Pore size	1	0.702 (0.137–3.592)	10.64 (1.269–89.23) **
Fitzpatrick skin type ^d^	1	0.788 (0.178–3.494)	1.908 (0.396–9.201)

PRS was generated by summing the number of the risk alleles in the collagen type XVII alpha 1 chain (*COL17A1*) rs805698, epidermal growth factor receptor (*EGFR*) rs1861003, and matrix metalloproteinase 16 (*MMP16*) rs6469206. ^a^ PRS ≤ 1, low PRS; 2–3 PRS, medium PRS; PRS ≥ 4, high PRS; ^b^ reference; ^c^ after adjusting for age, UV exposure, UV protection, menopause, sleep deprivation, morning tiredness, exercise, smoking, alcohol drinking, disease status, water intake, and vitamin supplementation; ^d^ scores of 4 and 5 in the Fitzpatrick skin type classification were combined; ** significantly different from the reference group at *p* < 0.01, and *** at *p* < 0.001.

**Table 6 ijerph-18-02044-t006:** Pearson correlation coefficients of skin evaluation scores with wrinkles and pore size.

	Wrinkles	Pore Size
Skin wrinkles	1.0	0.609 ***
Reflection of skin surface	0.155	0.112
Melanin deposition	0.398 ***	0.436 ***
Brown skin color	0.285 **	0.235 *
Blushing	0.099	0.105
Gloss	−0.042	0.189
Pore size	0.609 ***	1.0
Fitzpatrick skin type	0.352 ***	0.520 ***

* Significant correlation at *p* < 0.05, ** at *p* < 0.01, and *** at *p* < 0.001.

## Data Availability

The data presented in this study are available on request from the corresponding author.

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
