# Peer review of "Menopause, Ultraviolet Exposure, and Low Water Intake Potentially Interact with the Genetic Variants Related to Collagen Metabolism Involved in Skin Wrinkle Risk in Middle-Aged Women"

_ijerph, 2021, doi:10.3390/ijerph18042044_

Round 1

Reviewer 1 Report

This is an interesting study that is informative and novel. The link provided between different environmental factors and genetic biomarkers of aging-related to wrinkles is well-thought, however, the low number of participants in the study makes the interpretation of some of the data inconclusive. Besides, the association of skin wrinkling related to photoaging as a result of sun exposure has not been correctly addressed. In fact, the skin types of individuals (e.g. according to the Fitzpatrick classification) are the first determinant factor of the susceptibility of the skin to photodamage and wrinkling related to collagen and elastin breakdown. Therefore the authors must provide their data in liaison with the difference in skin type of the cohort and susceptibility to sun photodamage/photoaging.

The other parameters such as fruit and vitamin supplement intake seem well carried out. The water/liquid intake again is a parameter that can not be directly related to wrinkling of the skin, however, the degree of skin hydration could be a better parameter than liquid intake per se. This part must also be addressed by the authors, by providing the data for the cohort studied.    

Author Response

We appreciate the good comments on our paper. We carefully revised the manuscript, making a sincere effort to incorporate the reviewer and editor's suggestions according to each comment. We changed the manuscript in red text to be easily distinguished.

Review 1

This is an interesting study that is informative and novel. The link provided between different environmental factors and genetic biomarkers of aging-related to wrinkles is well-thought, however, the low number of participants in the study makes the interpretation of some of the data inconclusive.

: We agreed with the reviewer, and we wrote a small sample size as a limitation in the limitation section. However, the sample size was sufficient to achieve significance at α=0.05 and β=0.15 in the logistic analysis, which was estimated using the G power calculator. We suggested the further study is needed to be elucidated their association.

Besides, the association of skin wrinkling related to photoaging as a result of sun exposure has not been correctly addressed.

: We changed it in the Introduction section, line 55-61.

In fact, the skin types of individuals (e.g. according to the Fitzpatrick classification) are the first determinant factor of the susceptibility of the skin to photodamage and wrinkling related to collagen and elastin breakdown. Therefore, the authors must provide their data in liaison with the difference in skin type of the cohort and susceptibility to sun photodamage/photoaging.

: We provided the Fitzpatrick skin type of the participants in Table 2, and it was included for further analysis.

The other parameters such as fruit and vitamin supplement intake seem well carried out. The water/liquid intake again is a parameter that cannot be directly related to wrinkling of the skin, however, the degree of skin hydration could be a better parameter than liquid intake per se. This part must also be addressed by the authors, by providing the data for the cohort studied.    

: We understand the point of the reviewer. However, we did not measure the degree of skin hydration, and we want to indirectly adjust the degree of hydration with water and liquid intake. We added the point in the limitation.

Reviewer 2 Report

This paper by Park et al aims to find association between PRS and wrinkles and how social behaviors might influence this risk. The paper is well written and figures are appropriate, however, there is a need to strengthen the analysis and discussion of this paper, here are my recommendations:

A more complete data on other SNPs and their association is needed

The discussion need to be toned down, there is no evidence of interaction just association, this needs to be changed throughout the text including the title

There should be analysis of patients with elevated PRS that have contrasting demographic characteristics such as UV exposure, sunscreen application, water intake etc

Author Response

Review 2

This paper by Park et al aims to find association between PRS and wrinkles and how social behaviors might influence this risk. The paper is well written and figures are appropriate, however, there is a need to strengthen the analysis and discussion of this paper, here are my recommendations:

A more complete data on other SNPs and their association is needed.

: We added some significant SNPs related to oxidative stress and inflammation in Table 3.

The discussion needs to be toned down, there is no evidence of interaction just association, this needs to be changed throughout the text including the title

: Title was modified, and the discussion was toned down.

There should be analysis of patients with elevated PRS that have contrasting demographic characteristics such as UV exposure, sunscreen application, water intake etc.

: We adjusted these factors as covariates in odds ratio and means analysis. We also included the means and frequencies of the demographic characteristics. They did not have significant differences among the PRS groups.

Reviewer 3 Report

In the present manuscript, Park et al. investigated the skin wrinkle risk, including menopause, UV exposure and low water intake in middle-aged women, entitled “Menopause, ultraviolet exposure, and low water intake interact with the genetic variants related to collagen metabolism involved in skin wrinkle risk in middle-aged women”. Topic is interesting, but the present manuscript has some weak points as followed;

1. Please check the abbreviations through the article including Line 19, 22-23, 138 etc.

2. Please unify the term such as “use of UV blockers” and “usage of sunscreen”.

3. It is easy to understand if authors present the survey questionaires in a table.

4. Please combine the figures (Fig. 1A to 1C) into one.

Based on these reviews, the present study was suitable for publication in International Journal of Environmental Research and Public Health.

Author Response

We appreciate the good comments on our paper. We carefully revised the manuscript, making a sincere effort to incorporate the reviewer and editor's suggestions according to each comment. We changed the manuscript in red text to be easily distinguished.

Review 3

In the present manuscript, Park et al. investigated the skin wrinkle risk, including menopause, UV exposure and low water intake in middle-aged women, entitled “Menopause, ultraviolet exposure, and low water intake interact with the genetic variants related to collagen metabolism involved in skin wrinkle risk in middle-aged women”. Topic is interesting, but the present manuscript has some weak points as followed;

  1. Please check the abbreviations through the article including Line 19, 22-23, 138 etc.

: We checked all the abbreviations to be correct and consistent.

  1. Please unify the term such as “use of UV blockers” and “usage of sunscreen”.

: We changed the UV blocker into sunscreen.

  1. It is easy to understand if authors present the survey questionaires in a table.

: We added survey questionnaires in Table 1.

  1. Please combine the figures (Fig. 1A to 1C) into one.

 : Fig. 1A, 1B, and 1 C were combined as one.

Based on these reviews, the present study was suitable for publication in International Journal of Environmental Research and Public Health.